# Metabolic Innovations Underpinning the Origin and Diversification of the Diatom Chloroplast

**DOI:** 10.3390/biom9080322

**Published:** 2019-07-30

**Authors:** Tomomi Nonoyama, Elena Kazamia, Hermanus Nawaly, Xia Gao, Yoshinori Tsuji, Yusuke Matsuda, Chris Bowler, Tsuyoshi Tanaka, Richard G. Dorrell

**Affiliations:** 1Institut de Biologie de l’ENS (IBENS), Département de Biologie, École Normale Supérieure, CNRS, INSERM, Université PSL, 75005 Paris, France; 2Division of Biotechnology and Life Science, Institute of Engineering, Tokyo University of Agriculture and Technology, 2-24-16, Naka-cho, Koganei, Tokyo 184-8588, Japan; 3Department of Bioscience, Kwansei Gakuin University, Sanda, Hyogo 669-1337, Japan; 4Graduate School of Biostudies, Kyoto University, Kyoto 606-8501, Japan

**Keywords:** MMETSP, stramenopiles, biotechnology, secondary endosymbiosis, isoprenoids, plastid proteome

## Abstract

Of all the eukaryotic algal groups, diatoms make the most substantial contributions to photosynthesis in the contemporary ocean. Understanding the biological innovations that have occurred in the diatom chloroplast may provide us with explanations to the ecological success of this lineage and clues as to how best to exploit the biology of these organisms for biotechnology. In this paper, we use multi-species transcriptome datasets to compare chloroplast metabolism pathways in diatoms to other algal lineages. We identify possible diatom-specific innovations in chloroplast metabolism, including the completion of tocopherol synthesis via a chloroplast-targeted tocopherol cyclase, a complete chloroplast ornithine cycle, and chloroplast-targeted proteins involved in iron acquisition and CO_2_ concentration not shared between diatoms and their closest relatives in the stramenopiles. We additionally present a detailed investigation of the chloroplast metabolism of the oil-producing diatom *Fistulifera solaris*, which is of industrial interest for biofuel production. These include modified amino acid and pyruvate hub metabolism that might enhance acetyl-coA production for chloroplast lipid biosynthesis and the presence of a chloroplast-localised squalene synthesis pathway unknown in other diatoms. Our data provides valuable insights into the biological adaptations underpinning an ecologically critical lineage, and how chloroplast metabolism can change even at a species level in extant algae.

## 1. Diatoms: Powerhouses and Bellweathers in the Contemporary Ocean

Against the backdrop of rising atmospheric carbon dioxide concentrations and anthropogenic climate change, it is increasingly important to understand how photosynthesis functions in the ocean [1]. Among the diverse photosynthetic marine organisms in the contemporary ocean, diatoms are the most prominent and are estimated to be directly responsible for 40% total marine photosynthesis ([2] and references therein). In particular, diatoms dominate in polar waters and provinces that are chronically low in iron [3] and are the main photosynthetic producers in waters classified as ‘high in nutrients, low in chlorophyll’ (HNLC), notably large swathes of the Southern Ocean, equatorial Pacific Ocean, and north Pacific Ocean. However, they are not obligately oligotrophic species and bloom seasonally when nutrient or physical conditions change. This occurs mostly in coastal regions, following seasonal thermocline breaks, nutrient influx from the land, or aeolean fertilization. 

The environmental abundance of diatoms is in contrast to other related algal groups [4,5]. This may reflect underlying physiological innovations in diatoms, which are better able to tolerate physiological stresses, such as abnormal light regimes [6,7] and carbon dioxide limitation [6], than other algal lineages. In addition, diatoms display superior abilities to capture and utilise nutrients, including nitrogen [8] and iron [9,10]. Conversely, diatom abundance is being adversely affected by global environmental changes that reduce nutrient availability in the ocean. For example, in high latitude environments in which diatoms currently flourish, increased freshwater input from melting ice caps will depress nutrient concentration, likely favouring species with smaller cell sizes than diatoms [11]. Ocean acidification may also adversely affect nutrient acquisition and photoprotection in diatoms, although this remains debated [9]. 

Alongside their environmental abundance, the diatom group includes various phenotypes with possible agricultural and industrial applications, many of which may be lineage- or even species-specific. Diatoms can be applied as aquafeeds for fish because they contain suitable nutrients, and particularly lipid profiles such as eicosapentaenoic acid (EPA) and docosahexaenoic acid (DHA) [12,13]. Prior field trials with *Phaeodactylum, Nanofrustulum*, and *Navicula* have demonstrated the success of up to 5–10% replacement of conventional fishmeal with diatom stocks, and these may form the basis of next-generation ‘circular’ aquaculture techniques with increased capacity and reduced environmental impact [12,14]. 

Potential biotechnological applications of diatoms include the use of pigments (e.g., from *Phaeodactylum*) in cosmetics [15,16], diatom-derived polyunsaturated fatty acids as food supplements [17,18], diatom frustules (e.g., from benthic pennate diatoms) as UV-resistant coatings for photovoltaic cells [19], and even use in next-generation biomedical technologies, e.g., the use of diatom derived ice-binding proteins (from the polar species *Navicula glaciei*) in blood cryopreservation [20]. Much of current research is focused on oleaginous diatom species, which might be of particular interest for the production of algal biodiesel; for example, the raphid pennate species *Fistulifera solaris* is a can contain up to 60% of its dry mass as lipids [21], including high amounts of the valuable polyunsaturated fatty eicosapentenoic acid [22]. Techniques for mass cultivation [21,23] and transformation are established in some diatom species, including *F. solaris* and *Phaeodactylum tricornutum* [4,21,24], opening up significant windows for synthetic engineering and practical use of these species. 

Understanding diatom biology, and particularly which factors of their metabolism make them unique, may help us understand better how they will respond to environmental perturbation and offer new routes for the modification and exploitation of algal systems for industrial aquaculture. In this paper, we focus on which pathways delineate and optimise metabolic functions in the diatom chloroplast. To elucidate which of these processes have specifically contributed to the success of the diatom lineage, we reconstruct the probable metabolic functions contained in the diatom common ancestor, following its divergence from other algal lineages. We also explore specific biochemical adaptations associated with the chloroplasts of one diatom species, the biofuel producer *F. solaris*.

## 2. Taxonomic and Ecological Diversity of Diatoms

Diatoms arose within the stramenopile supergroup, which otherwise includes both non-photosynthetic and photosynthetic members [4,25]. Diatoms possess a silicon-rich cell wall called the frustule, which is an important trait for taxonomic classification [26,27]. Diatoms are classified into two major morphological categories (Figure 1A). The centric diatoms have radial symmetry, generally undergo anisogamous sexual reproduction, and contain multiple chloroplasts per cell [4,28] (Figure 1A). In contrast, the pennate diatoms have linear symmetry, typically have bigger frustules than centric diatoms [29], produce isomorphic gametes, and only have one chloroplast per cell (Figure 1A). Both centric and pennate diatoms are further divided into two sub-groups: the centrics into radial and polar types, based on descriptors of frustule symmetry, and the pennates into raphid and araphid types, based on the presence or absence, respectively, of a slit known as raphe at the centre of the frustule that facilitates motility on surfaces (Figure 1A).

Phylogenetic evidence places the radial centric diatoms as paraphyletic to all other lineages, the polar centric diatoms as paraphyletic to the pennate diatoms, the araphid pennates as paraphyletic to the raphid species, and the raphid pennates as monophyletic [36]. Fossil-calibrated molecular clock estimates place diatom origins in the Permian period (~320 Mya) with a shared common ancestor to another stramenopile group, the bolidophytes, at ~375 Mya [34]. Radial centric diatoms arose at least 250 million years before the present (Figure 1B), potentially placing them on a similar degree of evolutionary antiquity as flowering plants [34,37]. Polar centric and araphid pennate diatoms appear to have arisen in the Jurassic and Cretaceous (Figure 1B) [3,26], alongside falling atmospheric carbon dioxide concentrations [38]. Finally, the raphid pennate diatoms diversified in the early Oligocene (~40 Mya) (Figure 1B), alongside rising oceanic silicate concentrations, following the opening of the Drake Passage between South America and Antarctica [38]. 

Genome sequences have been completed for both centric (*Thalassiosira pseudonana, T. oceanica*) [39,40] and pennate diatom species (e.g., *Phaeodactylum*, *Fistulifera*) [21,24], revealing that these two distinct diatom morphogroups are more genetically distant to one another than humans are to pufferfish [24]. However, both diatom morphogroups are ecologically successful, with the most abundant genera (each comprising at least 4% of total diatom OTUs) in the *Tara* Oceans library including radial centric (*Corethron, Leptocylindrus*), polar centric (*Thalassiosira*), and pennate taxa (*Fragilariopsis, Pseudo-nitzschia*) [3]. Typically, centric diatoms are found predominantly in the open oceans, and pennate species are found in coastal and benthic waters [3,27]. By comparison, their immediate sister group within the stramenopiles, the bolidophytes, which do not ever form more 4% of the total photosynthetic abundance in any one *Tara* Oceans station [5].

## 3. Diatom Chloroplast Structure and Genomes

Diatoms and other photosynthetic members of the stramenopile algae possess a chloroplast derived from the secondary endosymbiosis of a red alga (Figure 2a). This chloroplast is closely related to the chloroplasts found in other lineages with secondary red chloroplasts (i.e., cryptomonads, haptophytes, and alveolates). Alternative origins have also been proposed, in particular a tertiary endosymbiotic origin, in which a red alga was acquired through secondary endosymbiosis through another algal group (e.g., cryptomonads [4,25]), which were in turn engulfed by the photosynthetic stramenopile ancestor (Figure 2b). The diatom chloroplast has a distinctive structure, consisting of an annular set of unstacked thylakoids around the periphery of the stroma (« girdle lamella »), enclosing a diffuse pyrenoid, and a ring-like genophore, and contains chlorophyll a, c1, and c2, as well as fucoxanthin and hexanoylfucoxanthin as typical pigments (Figure 2c) [4,41]. All of these features are shared with the bolidophytes and hence are not diatom-specific.

The diatom chloroplast retains a genome containing between 164 (*Astrosyne radiata*) and 204 genes (*Eunotia naegelii*) [4,46], with functions in photosynthesis, genome expression and protein import, cofactor (chlorophyll, thiamine, Fe-S cluster) biosynthesis, and a select number of conserved *ycf* genes of unknown function [46]. Some diatom species retain additional chloroplast-encoded functions, e.g., acetolactate synthase typically found in centric diatom chloroplast genomes [47], and a chloroplast-encoded light-independent protochorophyllide reductase so far known to be chloroplast-encoded solely in the polar centric diatom *Toxarium* [48]. 

Some diatoms, in particular within the genus *Nitzschia*, have lost the capacity to photosynthesize [49,50]. These species lack most chloroplast genes involved in photosynthesis and associated metabolisms, although intriguingly they retain components of a chloroplast ATP synthase. This complex has been proposed to function in the catabolism, rather than synthesis, of ATP, generating a trans-thylakoid proton gradient facilitating protein import through the *tat* translocase complex [50].

## 4. Import and Mosaic Origin of Diatom Nucleus-Encoded Chloroplast Proteins

As with other photosynthetic eukaryotes, the overwhelming majority of proteins that function in the diatom chloroplast are not encoded in its genome but are instead encoded in the nucleus and imported from the cytoplasm [25]. The diatom chloroplast, like those of other stramenopiles, haptophytes, and cryptomonads, is surrounded by four membranes which are, from inside out, the inner envelope membrane (iEM), the outer envelope membrane (oEM), the periplasmic membrane (PPM), and the chloroplast endoplasmic reticular membrane (cERM), which is contiguous with the endoplasmic reticulum (Figure 2c) [4]. 

Proteins encoded in the diatom nucleus are transported across each chloroplast membrane through individual transporter complexes, each of which recognises a different component of the chloroplast presequence. Diatom chloroplast-targeted proteins commence with an *N*-terminal signal peptide, recognised by a conventional ER import machinery in the cERM; followed by an aromatic amino acid or leucine, allowing recognition by a specialised protein import complex, « SELMA », that resides in the PPM; and finally a hydrophilic chloroplast transit peptide that allows import through the iEM and oEM [51,52]. 

Different proteins localise to different sub-compartments within this chloroplast, including the space between the PPM and oEM (the « perichloroplast compartment ») [53,54]; the pyrenoid and thylakoids [25,54]; and the intermembrane space between the iEM and oEM [10,25]. Other proteins may be trafficked into the chloroplast following glycosylation in the ER/Golgi body [55] or may be dual-targeted to the chloroplast and other compartments, such as the mitochondria [25,56]. 

A further level of complexity is contributed by the chimeric origins of the diatom chloroplast proteome (Figure 2B). Previously, we have characterised 770 nucleus-encoded and chloroplast-targeted proteins that are shared across, hence ancestral to, photosynthetic stramenopiles [4,25]. We noted that although 60% of the proteins with clearly attributable origins resolved with red algae, consistent with the endosymbiotic origin of the chloroplast, the remaining 40% originated from other sources [25]. These include a sizeable number of chloroplast-targeted proteins of green algal origin that are not found in red algae [25,57], alongside chloroplast-targeted proteins of bacterial origin, and proteins repurposed from other cellular compartments within the stramenopile host [25]. Most intriguingly, over 90 of the proteins identified in this study did not have obvious homologues (as inferred using BLAST top hit searches, with threshold evalue 1 × 10^−5^) outside of algae with secondary chloroplasts [25] and thus might have specifically evolved within this lineage [10,25].

## 5. Metabolic and Evolutionary Complexity of the Diatom Chloroplast 

In the time that diatoms have separated from related stramenopile groups, they have accumulated innovations that have allowed this group to rise to ecological prominence. This begs the question, how are diatom chloroplasts different to their nearest relatives with which they share ancestry? Here, we study stramenopile sequence libraries to infer innovations in the diatom chloroplast that may enable enhanced primary productivity, resistance to photochemical damage, and ability to effectively utilise key organic and inorganic nutrients in ocean environments [4]. We focus on four themes: iron metabolism, biosynthesis of organic metabolites, photoprotection, and CO_2_ concentrating mechanisms.

### 5.1. The Chloroplast Proteome of the Diatom Common Ancestor

We profiled the metabolic innovations that are likely to underpin the origins of the diatom chloroplast using methodology adapted from previous studies [25,58]. This involved subsets of chloroplast-targeted proteins from different stramenopile transcriptomes and genomes, identified via in silico prediction, for homologues of 9531 evolutionarily non-redundant chloroplast-targeted proteins identified in species with chloroplasts of secondary red algal origin via a composite BLAST pathway, based on reciprocal BLAST best-hit searches, using a floating *e*-value threshold. 

We mapped the distributions identified onto previously calculated diatom and stramenopile phylogenetic trees [4,25] to identify the most probable origin points of each protein, inferred to be the last common ancestor of all species in which the protein was detected. We also identified possible loss events, defined if a chloroplast-targeted protein was not detected in any members of the clade, the clade contained multiple species and at least one genome sequence, and the protein was detected in at least two successive sister-groups to the clade. A KEGG map of two salient time points—the common ancestor of pennate and polar centric diatoms and the common ancestor of all diatoms, bolidophytes, and hypogyristea (pelagophytes and dictyochophytes)—is provided in Figure 3A. A comparison of the proteins identified through this approach, via reciprocal BLAST best-hit analysis, to analogous chloroplast protein datasets from plants, red algae, and their closest relatives [59,60,61,62] is provided in Figure 3B. Complete outputs are provided in Appendix A.

The diatom chloroplast performs effectively all of the essential pathways identified in plant chloroplasts, including photosynthesis, central carbon and lipid metabolism, synthesis of plastidial amino acids (e.g., glutamate/ glutamine, cysteine, lysine, branched chain, and aromatic amino acids), chlorophyll and carotenoid synthesis, and essential plastid biogenesis pathways associated with expression of the chloroplast genome and protein import [25,58,61] (Figure 3A). However, it differs substantially in protein content from that of its ancestors, including (for example) substantial numbers of proteins exclusively shared with green algae and plants and not detectable in red algae and large numbers not present in primary chloroplast lineage (Figure 3B). Moreover, the diatom chloroplast is not identical to that of other stramenopiles, possessing a wider range of predicted metabolism pathways (Figure 3A), and as much as 1063 proteins not found in either the hypogyristea or stramenopile common ancestor (Figure 3B). 

Below, we outline key innovations in the chloroplasts of diatoms compared to close relatives within the stramenopiles, focusing on iron and nitrogen metabolism, photoprotection, and carbon dioxide acquisition. 

### 5.2. Iron Metabolism

Iron is essential for cellular life. Its flexible redox chemistry is at the heart of the fundamental metabolic processes of photosynthesis and nitrogen fixation, including three of the four major photosystem complexes (photosystem II, cytochrome b6/f, and in particular in photosystem I). The majority of iron in the contemporary ocean is in the insoluble form of Fe^3+^ complexed to oxyhydroxides, which is generally not as readily bioavailable as the more soluble Fe^2+^ or organically bound iron. The ecological success of diatoms is frequently linked to their unique and highly adapted iron physiologies, which match the chemical speciation of iron in the water column [4,36]. 

Diatoms may be able to better tolerate chronic iron deprivation by replacing iron-dependent proteins in chloroplast photosystems with iron-free alternatives (Figure 4A). These can include plastocyanin (which contains copper) instead of cytochrome c6 [64,65] and flavodoxin as an iron-free alternative to the photosystem I acceptor ferredoxin [66]. While plastocyanin genes are widespread across diatom species and are upregulated under Fe-limiting conditions [67,68], very few of the identified diatom plastocyanin sequences contain recognisable chloroplast-targeting peptides, and it remains to be confirmed where in the cell this protein functions [64]. Diatom flavodoxin genes, by contrast, are universally distributed, and many have chloroplast-targeting sequences. However, chloroplast-targeted flavodoxin protein at least arose in the common ancestor of diatoms, pelagophytes, and dictyochophytes (Figure 4A), rather than being an adaptation specifically confined to diatoms. 

Diatoms may additionally be able to tolerate iron limitation by efficiently mobilising iron, when it becomes available in the water column. This is evidenced during periods of transient enrichment of Fe, e.g., experimentally during artificial Fe fertilization in situ, which results in diatom-dominated blooms [4]. This effective iron acquisition is partly due to a novel pathway, allowing diatom chloroplasts to utilise iron complexed to siderophore molecules. Siderophores are iron-chelators, are typically biosynthesized by bacteria and fungi, comprise 99% of the organic iron pool, and were previously not considered accessible to photosynthetic eukaryotes [70]. Siderophore-bound iron is taken up from the diatom cell surface via an endocytosis pathway mediated by ISIP1, an ‘iron-starvation induced protein’ [10,71]. The ISIP1-siderophore complex is then trafficked toward the chloroplast, and possibly into the PPC, where reduction occurs, releasing iron from its organic chelator [10,72]. Notably, ISIP1 does not have conventional chloroplast targeting peptide, and the exact mechanism by which it is transported from the plasma membrane to this compartment remains unknown (Figure 4A) [10]. However, ISIP1 is only found in diatoms, with a distantly divergent homologue found in some pelagophyte, haptophyte, and dinoflagellate libraries, suggesting that it may be a diatom-specific strategy to facilitate iron delivery to the chloroplast (Figure 4) [10]. 

Finally, diatom chloroplasts may possess large internal reserves of iron, allowing them to maintain a large reserve of iron resources and correct for temporal differences in iron availability. For example, the iron storage protein ferritin [10] is predicted to be chloroplast-targeted across a wide range of diatoms, and is absent from other stramenopiles, although chloroplast-targeted equivalents are detectable in haptophytes and cryptomonads (Figure 4A). What is unclear is the cellular function of ferritin in cells, as a storage protein (as evidenced for *Pseudo-nitzschia* [73]) or as a homeostatic buffer. We should stress it is unlikely to be the only storage protein or the only mechanism for homeostasis, as other whole-cell mechanisms for iron homeostasis have been proposed [72]. 

### 5.3. Nitrogen Metabolism

Recently, it was proposed that diatoms possess a chloroplast-targeted glutamine-ornithine cycle, allowing them to recycle chloroplastidial glutamine to ornithine, which could then be exchanged with the cytoplasm [4]. This is particularly intriguing as it complements the finding that diatoms possess a mitochondria-targeted urea cycle, which would conversely allow them to synthesize glutamine in the mitochondria from ornithine [74]. Working in parallel, the organelles can sensitively equilibrate the availabilities of free amino acids, achieving a level of cellular integration not observed in other species [8,75]. Crucially, enzymes in this pathway are upregulated under nitrogen limitation and are coregulated with chloroplast-targeted nitrate reductase, suggesting roles in mediating nitrate assimilation [8,74,76]. 

We have localised all four of the involved enzymes in this cycle to the chloroplast of *P. tricornutum*, either through the localisation of full-length GFP constructs (acetylornithine transaminase, acetylglutamate kinase, glutamate N-acetyltransferase) or N-terminal GFP constructs (N-acetyl-δ-glutamyl-phosphate reductase; Appendix AA), following previous methodology [25]. Chloroplast-targeted copies of all four enzymes can be detected in other stramenopiles, cryptomonads, and haptophytes, although the complete cycle of four enzymes appears to only be chloroplast-targeted in diatoms (Figure 4B). Understanding the exact function of this pathway will depend on the characterisation of mutant lines for each protein.

### 5.4. Photoprotection

Diatoms possess a number of photo-physiological strategies for tolerating excess light that are shared with green algae, but not with the red antecedents of their chloroplasts [76]. These include an inferred xanthophyll cycle, which allows diatoms to effect non-photochemical quenching through the conversion of the accessory pigment diadinoxanthin into diatoxanthin and an expanded set of LHCx/Li818 family light harvesting proteins, which are implicated in the tolerance of light stress, not least by inducing a functional xanthophyll cycle [77]. The exact evolutionary origin of these pathways, including whether they can be truly considered as absent from red algae, has been considered elsewhere [78]. However, the effector enzymes of the xanthophyll cycle (violaxanthin de-epoxidase and zeaxanthin epoxidase), and LHCx/Li818 family proteins, are broadly conserved across algae with secondary red algal chloroplasts and are therefore not a diatom-specific innovation [57,79].

Recently, an additional photoprotective strategy has been evidenced in diatoms, in which excess chloroplast reducing potential is translocated into the mitochondria and dissipated through respiratory electron transport [25,79]. Excess mitochondrial ATP might conversely be imported into the chloroplast and used to activate ATP-dependent metabolic strategies (such as the Calvin Cycle) in the absence of chloroplast ATP, e.g., following the dark-light transition of a seasonal or circadian light cycle [7,80]. This process has been documented in other lineages (e.g., the green model alga *Chlamydomonas*), but only in cases where alternative photochemical quenching strategies have diminished function [7,75]. Thus, diatoms depend on this metabolite exchange much more substantially than other lineages of algae. Ultimately, identifying which effector proteins are involved in this process may provide clues into whether it is a diatom-specific or more widespread chloroplast evolutionary adaptation.

Finally, we have uncovered evidence for a diatom-specific chloroplast antioxidant strategy. We note that, in our KEGG map of chloroplast functions, the last common ancestor of the pennate and polar centric diatom lineages (Figure 3A, circled) is predicted to possess a complete chloroplast-targeted tocopherol (vitamin E) biosynthesis pathway. In contrast, other stramenopile algae lack a chloroplast tocopherol cyclase, with one possible exception in dictyochophytes (Figure 4C; Appendix A). Tocopherol is a known antioxidant, having overlapping functions to xanthophylls in scavenging of singlet oxygen and preventing the propagation of lipid peroxidation in plant chloroplasts [7,81]. Understanding the physiological significance of this pathway will depend first on identifying whether tocopherol is indeed present in diatom chloroplast preparations [82], and phenotyping mutants of the tocopherol cyclase, and/or its downstream methyltransferase (Figure 4C).

### 5.5. CO_2_ Concentrating Mechanisms (CCM)

Photosynthetic efficiency is a likely candidate trait for determining ecological success in marine environments. Minor adjustments that increase the proportion of fixed carbon could quickly translate to enhanced growth and standing biomass. In particular, carbon dioxide has limited solubility in water; this may therefore limit the carbon catalytic activity, and increase the relative proportion of wasteful oxygenase activity of RuBisCO in aquatic photosynthesis [32,83]. As a result, many photosynthetic algae utilise CO2 concentrating mechanisms that enrich carbon dioxide availability to RuBisCO. 

To date, it has been demonstrated that diatoms, including *Phaeodactylum tricornutum* and *Thalassiosira pseudonana*, have a biophysical CCM (using carbonic anhydrases and bicarbonate transporters) [54,83,84]. Some diatoms may additionally utilise biochemical strategies such as C4 photosynthesis [85], although an evident C4 operation has so far been detected solely in one centric species, *T. weissflogii* [83,85]. Previously, we and others have shown that diatoms possess a diverse set of carbonic anhydrases, at least some of which localise to different chloroplast sub-compartments [86], and which may confer an advantage in dynamic open ocean environments. These enzymes include an unusual, theta-class carbonic anhydrase, which in *P. tricornutum* localises within the lumen of pyrenoid-penetrating thlaykoid membranes and is directly implicated in the release of CO2 into chloroplast RuBisCO [54]. 

To gain a sense of how these proteins evolved, we profiled the distribution of all experimentally characterised carbonic anhydrases from *P. tricornutum* and *T. pseudonana* across stramenopile, haptophyte and cryptomonad genomes and transcriptomes, as above (Appendix A; Figure 5A) [54,86]. Through this approach, we identified 434 chloroplast-targeted theta-carbonic anhydrases in 48 of the 86 species studied, including the early-diverging radial centric genus *Leptocylindrus* (Appendix A). This implicates chloroplast theta-carbonic anhydrases as being present in the diatom common ancestor. We observed much more restricted evolutionary distributions for chloroplast-targeted isoforms of other diatom chloroplast carbonic anhydrases, including the *Phaeodactylum* pyrenoidal beta-carbonic anhydrases PtCA1 and PtCA2 (Phatr3_45443, 51305), and the *Thalassiosira* stromal alpha carbonic anhydrase TpCA1 [54,86,87,88] (Appendix A). Although we could identify chloroplast-targeted theta-carbonic anhydrases in other stramenopile, cryptomonad, and haptophyte lineages, we noted that diatom theta-carbonic anhydrases contained a much larger proportion of chloroplast-targeted proteins than orthologues identified in other lineages, suggesting this predominance is a specific innovation within the diatom chloroplast (Figure 5A).

A consensus Bayesian and RAxML tree of theta carbonic anhydrases revealed three distinct groups of diatom chloroplast proteins (Figure 5B). Two of the clades (clades 1 and 2) possess immediate sister-groups within the stramenopiles (respectively, pelagophytes and bolidophytes), suggesting probable vertical inheritances, although the sister-group sequences appear not to be chloroplast-targeted (Figure 5). The final clade of diatom chloroplast theta carbonic anhydrases (clade 3) resolves within a paraphyletic clade of haptophyte algae, suggesting a probable horizontal transfer event from haptophytes into the diatom common ancestor. At least some of the haptophyte sequences have chloroplast-targeting sequences, so it is possible that this anhydrase already possessed chloroplast functions prior to its acquisition by the diatom ancestor (Figure 5A,B). Overall, our data is consistent with theta carbonic anhydrases being recruited through horizontal gene transfer and the recycling of enzymes from other cellular compartments in the diatom ancestor.

Finally, using similar methodology, we considered the distribution and inferred subcellular localisation of the new, iota-class of carbonic anhydrases, recently reported to function in the *T. pseudonana* chloroplast (Figure 5A and Appendix A) [89]. We noted that chloroplast-targeted copies of this enzyme are widespread across diatoms and are present in extremely high numbers in certain species, including 10 or more in *F. solaris, Odontella aurita,* and *Ditylum brightwellii* (Appendix A). We could not detect chloroplast-targeted copies of this enzyme in the genus *Proboscia,* and a subclade of *Chaetoceros* (sp., UNC1202, *curvisetus, brevis*) from which chloroplast-targeted theta-carbonic anhydrases were also globally absent (Appendix A). In contrast to the situation for theta-carbonic anhydrases, there was no apparent bias in favour of chloroplast-targeted proteins in diatoms compared to other stramenopile, haptophyte or cryptomonad groups, with a greater proportion of pelagophyte and dictyochophyte sequences possessing chloroplast-targeting peptides (Figure 5A; Appendix A), suggesting it is an evolutionary innovation that precedes the origins of the diatom chloroplast. 

## 6. Industrial Futures: Remodelling Chloroplast Metabolism in the Oil-Producing Diatom *Fistulifera solaris*

Diatoms show diverse phenotypes, and some species produce valuable products for industrial application. For instance, the raphid pennate diatom *F. solaris* accumulates large quantities of lipids, which is of interest for conversion to biodiesel (Figure 6A) [21]. Since photosynthetic eukaryotes including diatoms typically synthesize glycerolipids in the chloroplast and ER [21], *F. solaris* may have unique alterations to its chloroplast metabolism to facilitate lipid production. 

### 6.1. Remodelling of Lipid and Amino Acid Metabolism in the F. solaris Chloroplast

Using the methodology described above, we compared the complement of nucleus-encoded and chloroplast-targeted proteins inferred from the *F. solaris* genome to those of other diatom species. We noted distinct differences between *F. solaris* and other diatoms. These include the probable absence of a chloroplast-targeted pyruvate carboxylase (PC), pyruvate phosphate dikinase (PPDK), and phosphoenolpyruvate synthase (PEPS), which catalyze the conversion of pyruvate to phosphoenolpyruvate (Figure 6(Bi,Ci)) (Appendix A) and are otherwise widespread across diatoms. 

Conversely, only *F. solaris* among diatoms has a chloroplast-targeted pyruvate/2-oxoglutarate dehydrogenase complex (PDC), which generates acetyl-CoA, a substrate for fatty acid synthesis (Figure 6(Bi,Ci)) (Appendix A). We confirmed that the *F. solaris* PDC protein transit peptide cleavage site is upstream of the conserved PDC domain (Appendix A) and is therefore unlikely to be an annotation artefact or internal region of a non-chloroplast-targeted protein. The chloroplast-targeted PDC might participate in supplying acetyl CoA for chloroplast fatty acid synthesis, as has previously been proposed to occur in higher plants and *C. reinhardtii* [90], contributing to its oleaginous phenotype.

We found a possible remodeling of branched-chain amino acid (BCAA) synthesis pathways, which convert 2-oxybutyrate and pyruvate into isoleucine, valine, and leucine, in *F. solaris*. Unlike *P. tricornutum* and *T. pseudonana*, no chloroplast-targeted homologs of acetohydroxyacid isomeroreductase (AHAIR) and branched-chain amino acid transferase (BCAT) were identified in *F. solaris* (Figure 6(Bii,6Cii)) (Appendix A). We could identify equivalents of these enzymes functioning in the mitochondria (BACT) and the cytoplasm (AHAIR), suggesting that these essential amino acids are synthesized elsewhere in the *F. solaris* cell. 

The absence of chloroplast-targeted copies of each enzyme might relate to the oleaginous phenotype of *F. solaris,* as both are related to NADPH metabolism. AHAIR directly utilises NADPH as a reducing agent, and BCAT requires NADPH to regenerate its substrate glutamate from 2-oxoglutarate by glutamate synthase [91]. The absence of these pathways from the *F. solaris* chloroplast might indirectly increase NADPH availability for alternative pathways, such as chloroplast fatty acid synthesis. It has previously been shown that both the downregulation of plastidial amino acid biosynthesis and overexpression of NADPH-producing enzymes in diatoms accelerates neutral lipid accumulation in diatoms [92,93]. Ultimately, the absence of these enzymes, and the phenotypic consequences for the *F. solaris* chloroplast, awaits experimental verification.

### 6.2. Modified Chloroplast Isoprenoid Synthesis in Specific Diatoms

Unlike other diatoms, *F. solaris* encodes a chloroplast-targeted squalene synthase in its nuclear genome (Figure 6(Biii,Ciii)) (Appendix A). Squalene is synthesized from farnesyl and geranylgeranyl diphosphate, which are converted by squalene synthase (SQS) and farnesyl diphosphate synthase (FPPS), respectively. As per the situation in most other diatoms [92], SQS in *F. solaris* is fused with isopentenyl diphosphate isomerase (IDI) (Figure 6Ciii). We confirmed that the *F. solaris* FPPS target sequence was positioned upstream of the conserved domain, indicating that it is genuinely chloroplast-targeted (Appendix A). We localized SQS in *F. solaris* using a full-length intron-containing C-terminal fusion GFP overexpression construct, and it localized in the vicinity of the chloroplast (Figure 6D). This localization pattern is similar to PPM proteins in *P. tricornutum* [94].

Elsewhere within our dataset, we identified chloroplast-targeted isoprenoid biosynthesis enzymes in *Pseudo-nitzschia* sp., *Synedra acus*, and *Rhizosolenia setigera* (Figure 4, Figure 6Ciii); chloroplast FPP synthesis has also been predicted in *Haslea ostrearia*. By contrast, plant FPPS are found to localise in the cytosol, mitochondria, peroxisome, and possibly the chloroplast [98], while SQS localizes to the ER membrane in *Arabidopsis thaliana* [99]. Squalene production has been documented in other microalgae (e.g., the green alga *Botryococcus braunii* [100]), although the localisations of the enzymes involved remain unknown. Notably, the enzymes contributing to these metabolic fluxes vary between each species, with only *F. solaris* possessing a chloroplast-targeted IDI-SQS and FPPS, compared to, e.g., a chloroplast-targeted IDI in *R. setigera*. Thus, a complex chloroplast isoprenoid metabolism may have evolved in parallel in different diatoms.

The exact function of these enzymes in diatom chloroplasts remains to be determined. The enzymatic activity of isopentenyl diphosphate isomerase can be compensated by another enzyme, hydroxymethylbillance-CoA reductase, which is chloroplast-targeted across the stramenopiles, so it is therefore not essential for carotenoid synthesis [25,100]. However, the presence of chloroplast-targeted IDI-SQS and FPPS might facilitate the production of secondary metabolites that would allow diatoms to survive diverse biotic and abiotic pressures, similar to the functions of polyprenoid biosynthesis in plants [77] Isopentenyl diphosphate synthase activity in *R. setigera* has been proposed to be involved in the synthesis of highly branched isoprenoids, which may function in cryoprotection 92, and may include the diatom-specific isoprenoid IP25, which can be used as a biomarker for primary production in sea ice [92,101]. The adoption of different polyprenoid biosynthesis pathways in different diatoms might therefore confer different functions to these species.

## 7. Concluding Remarks

In this review, we discussed the mosaic origins, complex structure, and metabolic diversity of the diatom chloroplast. Focusing on innovations that might help explain the success of diatoms in the contemporary ocean, we have noted that diatoms possess unique chloroplast features, specifically within iron metabolism, biosynthesis of organic metabolites, photoprotection, and CO2 concentrating mechanisms. However, there are many other chloroplast proteins that may be unique to diatoms, with limited structural homology to other annotated proteins [25]. Characterisation of mutant lines of these proteins remains imperative to understanding their function. Environmental sampling projects (e.g., the *Tara* Oceans Survey [3]) may be instrumental in allowing us to deduce transcriptional patterns and the environmental relevance of understudied diatom proteins in current and future climates.

Comparing the chloroplast-targeted proteins of species with industrially desirable phenotypes, such as *F. solaris*, may reveal candidate pathways for engineering in other organisms. For example, it will be interesting to determine if suppressing chloroplast pyruvate or BCAA metabolism in *P. tricornutum* can enhance lipid production, which would support a link between these proteins and the oleaginous phenotype of *F. solaris*. Similarly, heterologous expression of chloroplast-targeted proteins aimed at modifying isoprenoid metabolism in *F. solaris* and other diatoms may reveal the function of these enzymes and might also allow us to generate new and industrially relevant metabolites in easily cultivable species.

Overall, the unique chloroplast metabolic pathways identified within multi-species transcriptome datasets help us understand more clearly the features underpinning diatom biology.

## Figures and Tables

**Figure 1 biomolecules-09-00322-f001:**
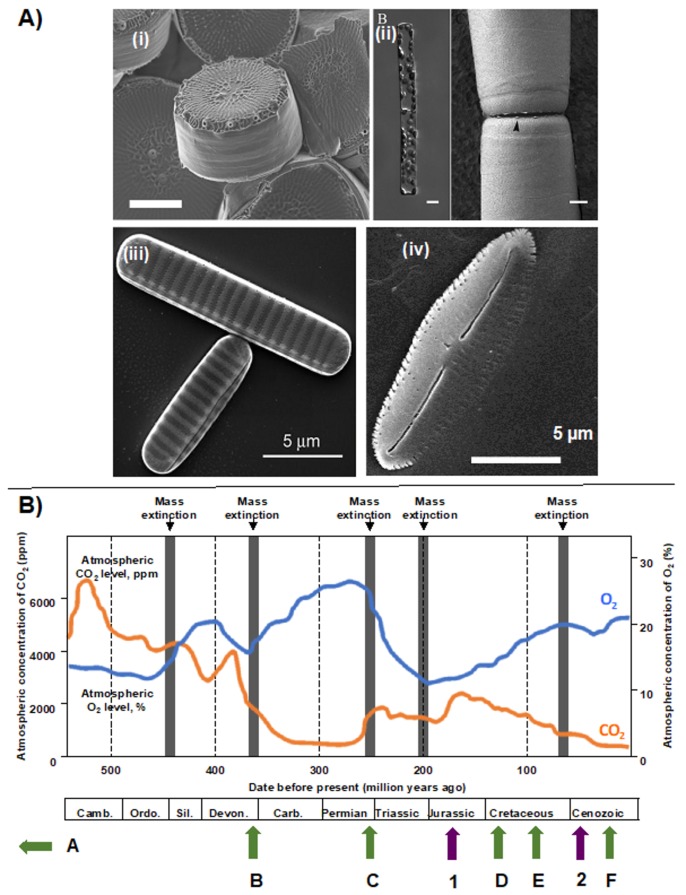
Diatom diversity and origins. Panel (**A**) shows scanning electron micrograph images of (**i**) *Thalassiosira pseudonana* (polar centric diatom, scale bar: 2 µm; [30]); (**ii**) *Leptocylindrus hargravesii* (radial centric diatom, scale bar: 5 µm); (**iii**) *Fragilariopsis cylindrus* (raphid pennate diatom, [31]); (**iv**) *Fistulifera solaris* JPCC DA0580 (raphid pennate diatom, authors’ own image). Panel (**B**) shows a global timeline of atmospheric CO_2_ (orange) and O_2_ concentrations (blue), adapted from [32], with key events in diatom evolution, as described in [33,34,35]. Abbreviated geological epochs are as follows: Camb., Cambrian; Ordo., Ordovician; Sil., Silurian; Dev., Devonian; Carb., Carboniferous. Green arrows show key points in diatom evolution: (**A**), median inferred radiation from molecular clock data of photosynthetic stramenopiles (>600 MYA); (**B**), median inferred date from molecular clock data for divergence of diatoms and bolidophytes (350 MYA); (**C**), earliest identifiable diatom fossils (250 MYA); (**D**), earliest identifiable pennate diatom fossils (125 MYA); (**E**), earliest identifiable polar centric diatom fossils (95 MYA); (**F**), earliest identifiable raphid pennate diatom fossils (20 MYA). Purple arrows show corresponding events in geological history that may have impacted diatom evolution and diversification: **1**, subduction of Tethyan trench (150 MYA); **2**, uplift of Himalayan plateau and opening of Drake passage and Southern circumpolar current (between 55 and 41 MYA). Figure 1Ai: Reproduced with permission from Eike Brunner, Patrick Richthammer, Hermann Ehrlich, Silvia Paasch, Paul Simon, Susanne Ueberlein, Karl-Heinz van Pée, Angewandte Chemie International Edition; published by John Wiley and Sons, 2009. Figure 1Aii: Reproduced with permission from Deepak Nanjappa, Wiebe H. C. F. Kooistra, Adriana Zingone, Journal of Phycology; published by John Wileyand Sons, 2013. Figure 1Aiii: Reproduced under the Creative Commons license from Thomas Mock, Robert P. Otillar, Jan Strauss, Mark McMullan, Pirita Paajanen, Jeremy Schmutz, Asaf Salamov, Remo Sanges, Andrew Toseland, Ben J. Ward, Andrew E. Allen, Christopher L. Dupont, Stephan Frickenhaus, Florian Maumus, Alaguraj Veluchamy, Taoyang Wu, Kerrie W. Barry, Angela Falciatore, Maria I. Ferrante, Antonio E. Fortunato, Gernot Glöckner, Ansgar Gruber, Rachel Hipkin, Michael G. Janech, Peter G. Kroth, Florian Leese, Erika A. Lindquist, Barbara R. Lyon, Joel Martin, Christoph Mayer, Micaela Parker, Hadi Quesneville, James A. Raymond, Christiane Uhlig, Ruben E. Valas, Klaus U. Valentin, Alexandra Z. Worden, E. Virginia Armbrust, Matthew D. Clark, Chris Bowler, Beverley R. Green, Vincent Moulton, Cock van Oosterhout, Igor V. Grigoriev, Nature; published by Springer Nature Publishing AG, 2017. Figure 1B: Reproduced with permission from Richard G. Dorrell, Alison G. Smith, Eukaryotic Cell; published by American Society for Microbiology, 2011.

**Figure 2 biomolecules-09-00322-f002:**
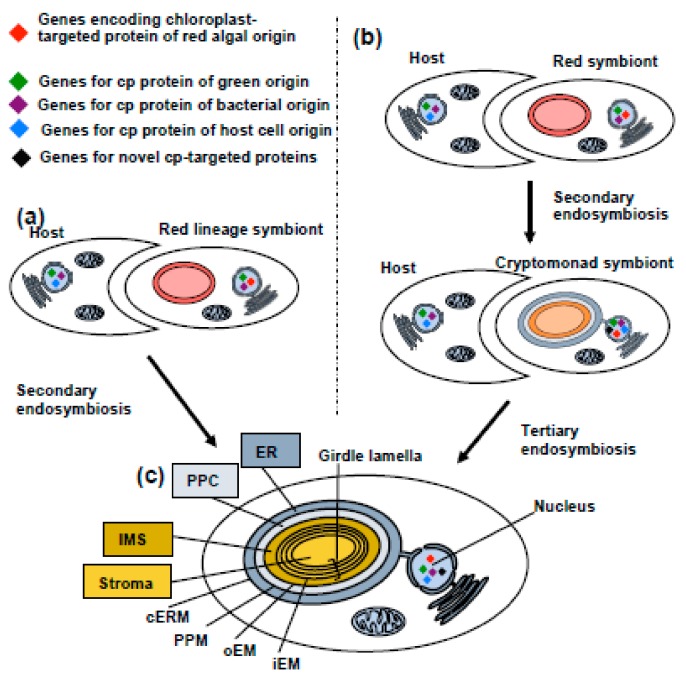
Origins and structure of the diatom chloroplast. This schematic figure shows two alternative hypotheses for the origins of the diatom chloroplast [4,25,42]: (**a**) secondary endosymbiosis of a red alga by a common ancestor of photosynthetic stramenopiles or (**b**) tertiary endosymbiosis of a cryptomonad-like organism, itself harbouring a chloroplast of secondary, red algal endosymbiotic origin. Either the host or the endosymbionts may have possessed genes retained from a cryptic endosymbiont of green algal origin [25,43,44,45], although this remains debated [44,45]. Other chloroplast-targeted proteins (indicated by the presence of coloured diamonds) may have been recruited from bacterial sources, either in the host or symbiont, proteins or paralogous copies of proteins previously targeted to other host cell organelles or may even have evolved de novo at the point of endosymbiosis [25]. (**c**) shows a schematic diagram of the four membranes surrounding the diatom chloroplast, adapted from [4]. Abbreviations are as follows: cERM; chloroplast endoplasmic reticular membrane; ER, endoplasmic reticulum; iEM, inner envelope membrane; IMS, intermembrane space; oEM, outer envelope membrane; PPC, periplastid compartment; PPM, periplastid membrane.

**Figure 3 biomolecules-09-00322-f003:**
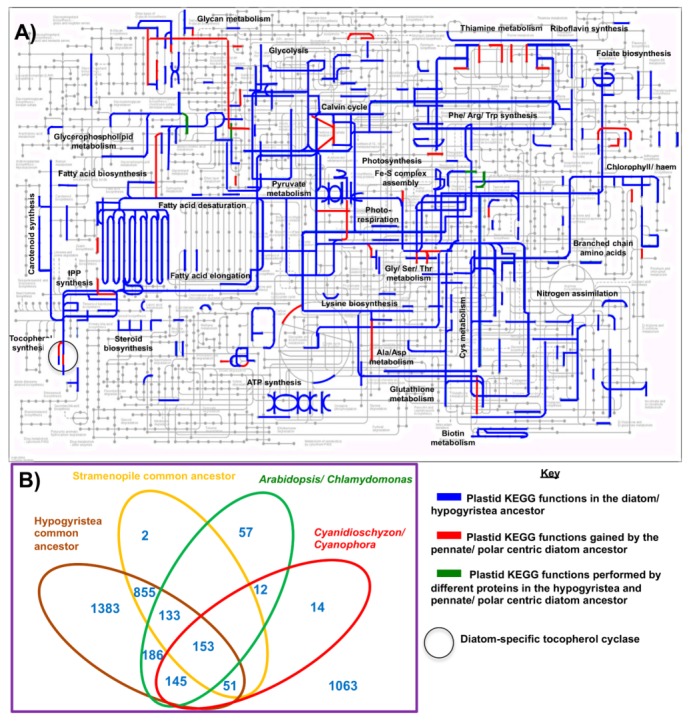
Innovations in metabolic complexity in the diatom chloroplast. Panel (**A**) shows KEGG (Kyoto Encyclopedia of Genes and Genomes) pathways that could be assigned to the chloroplasts of diatoms and their close relatives, based on the evolutionary distributions of 9531 chloroplast-targeted proteins identified by Dorrell et al. [25,58]. KEGG functions were identified using KEGG mapper [63], and key chloroplast metabolism pathways are annotated following [60]. Blue lines show metabolic pathways present in the common ancestor of diatoms, bolidophytes, pelagophytes, and dictyochophytes; red lines show pathways subsequently gained by a common ancestor of the chloroplasts of pennate and polar centric diatoms; and green lines show pathways conserved between but performed by different enzymes in each lineage. Panel (**B**) compares the protein sets identified in the ochrophyte common ancestor; the common ancestor of diatoms, bolidophytes, pelagophytes, and dictyochophytes; and the common ancestor of pennate and polar centric diatoms to published experimental and phylogenomic plastid protein datasets for *Arabidopsis* [60], the green alga *Chlamydomonas* [61], the red alga *Cyanidioschyzon* [62], and the glaucophyte *Cyanophora* [59]. Protein co-occurrence is detected by reciprocal BLASTp/BLASTp search best-hit between each dataset with bidirectional threshold evalue 1 × 10^−5^. Complete plastid proteome outputs for each species, and reciprocal BLAST best hit analysis between proteome datasets, are provided in Appendix A.

**Figure 4 biomolecules-09-00322-f004:**
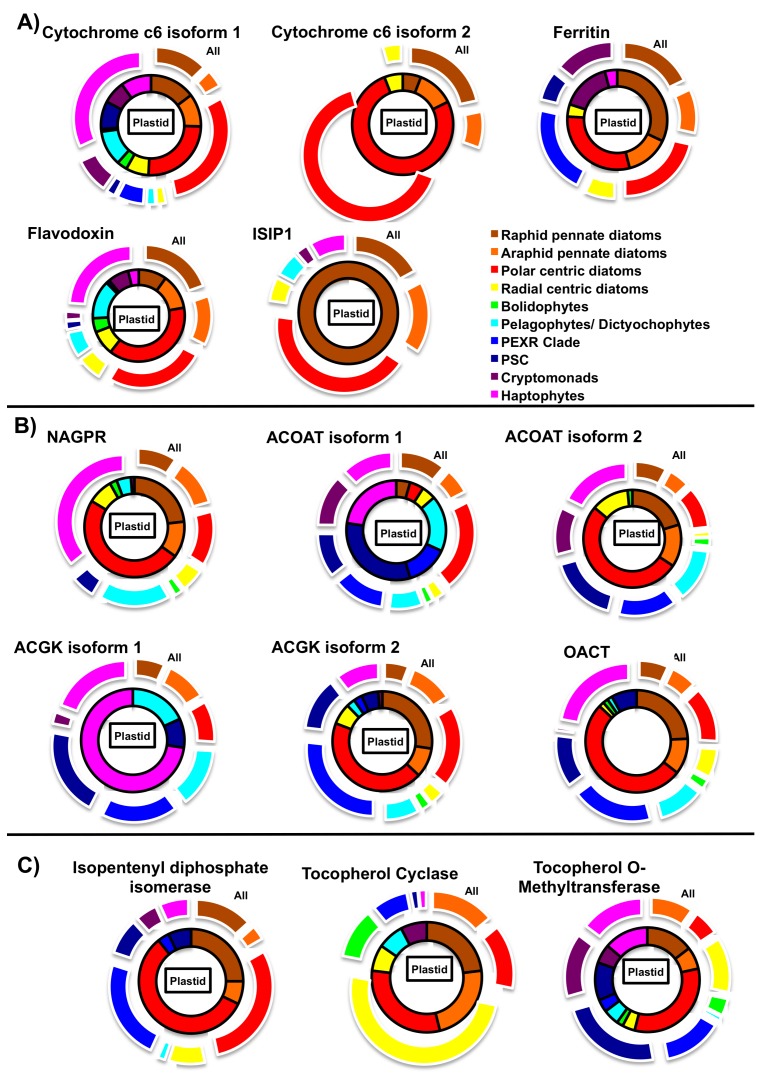
Evolutionary distributions of chloroplast proteins restricted to diatoms and their close relatives. This figure shows the evolutionary distributions of chloroplast-targeted proteins involved in (**A**) iron metabolism, (**B**) ornithine metabolism, and (**C**) organic compound synthesis, inferred from the pipeline used in Figure 3A, across photosynthetic stramenopiles, haptophytes, and cryptomonads. Lineages are arranged into monophyletic taxonomic groups, following published phylogenies [36,69]. The inner wheel of each diagram (bounded with black lines) shows the total numbers of species within a given lineage for which chloroplast-targeted proteins were identified, and the outer wheel of each diagram shows the number of species within a given lineage for any protein homologue (including those with non-chloroplast or ambiguous targeting signals) could be identified. Exemplar GFP localisations for enzymes associated with the probable diatom chloroplast ornithine cycle are shown in Appendix A. Presence and absence distributions of each protein, listed by species, are provided in Appendix A.

**Figure 5 biomolecules-09-00322-f005:**
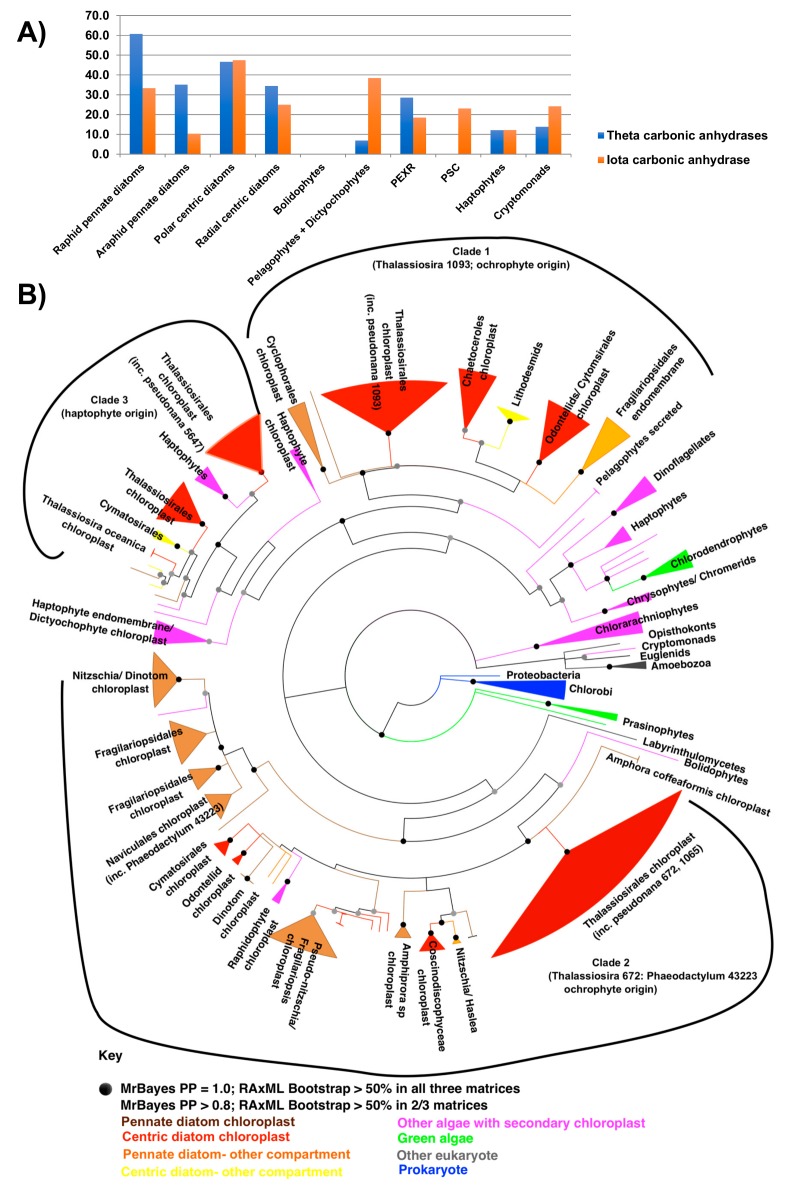
Evolution of chloroplast-targeted diatom carbonic anhydrases. Panel (**A**) shows the number (labels) and proportion of chloroplast-targeted sequences (bars) identified for theta and iota carbonic anhydrases in diatoms and other stramenopile, cryptomonad and haptophyte lineages, by reciprocal BLAST best hit analysis. Detailed outputs are provided for the distribution of theta, alpha, and beta carbonic anhydrases and iota carbonic anhydrases, across diatom species in Appendix A, respectively. Panel (**B**) shows the consensus Bayesian topology of a 204 taxa × 210 aa alignment of theta-carbonic anhydrase sequences across the tree of life. Three clades of diatom chloroplast-targeted proteins are shown with circular brackets. Sequence names are shaded by taxonomic origin and are labelled with predicted localisations (chloroplast, endomembrane system, and/or mitochondria). Alignment and tree topologies under each programme and matrix considered are shown in Appendix A.

**Figure 6 biomolecules-09-00322-f006:**
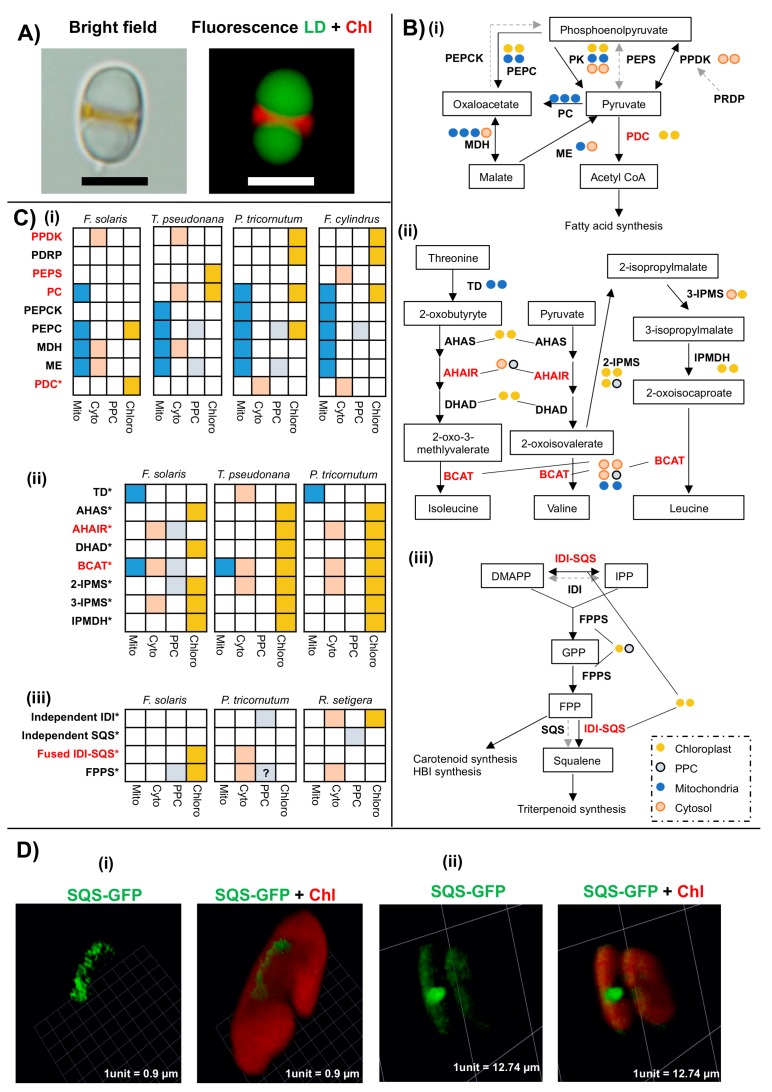
Innovations in the *Fistulifera solaris* chloroplast. Panel (**A**) shows a *Fistulifera solaris* cell visualized under bright field and fluorescence microscopy. The conspicuous lipid droplets (LD) are visualized using BODIPY (boron-dipyrromethene) 505/515 staining (in green) and chlorophyll fluorescence (Chl) is shown in red; scale bar = 5 µm. Panel (**B**) shows schematic pathway diagrams for (**i**) pyruvate hub metabolism; (**ii**) branched chain amino acid synthesis; and (**iii**) isoprenoid metabolism as inferred from the *F. solaris* genome [21]. Each circle equates to one distinct copy of each gene. Genes labelled in red are those for which *F. solaris* has different metabolic arrangements to other diatom species considered. Panel (**C**) shows detailed localisation predictions for homologues of key enzymes implicated in modified chloroplast metabolism in *F. solaris* [21], alongside equivalent data for (**i**) *T. pseudonana, P. tricornutum, and F. cylindrus* [94], (**ii**) *T. pseudonana* and *P. tricornutum* [91], and (**iii**) *P. tricornutum* and *R. setigera* [24,95]. The symbol ‘?’ depicts a sequence without a cleavable signal peptide but predicted as a transmembrane protein. Sequence IDs shown in this figure are provided in Appendix A. Panel (**D**) shows a three-dimensional fluorescence microscope image of a line of *F. solaris* expressing a C-terminal GFP full length fusion construct of native squalene synthase protein, following previous methodology [21,25,96]. Cells visualised show two localisation patterns, both previously indicated to the periplastid compartment [97]—either (**i**) elongated localisation along the chloroplast periphery or (**ii**) across the mid-band of dividing chloroplasts. A diagram of the targeting sequence of this protein is provided in Appendix A.

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
