# Peer review of "Metabolic Innovations Underpinning the Origin and Diversification of the Diatom Chloroplast"

_biomolecules, 2019, doi:10.3390/biom9080322_

Round 1
Reviewer 1 Report
The manuscript is very interesting and brings new values in the field of diatom chloroplasts and provides explanations regarding the ecological success of this group of organisms and tips on how best to use the biology of these organisms in biotechnology.
The work mentioned the possibility of using diatom in biotechnology, but this fragment was poorly developed without providing examples of species and practical application. Nothing has been mentioned about the use of diatoms in agriculture and their important role. In the context of work, these data should be supplemented.
The authors may consider the unification of units in Figure 1. This will facilitate the possibility of comparing the given values for CO2 and O2 content in%.
Fig. 3, 4 Descriptions in the drawings should be improved because they are of low quality.
The captions in fig. 5 are illegible. The quality of the descriptions should be improved.
Author Response
The manuscript is very interesting and brings new values in the field of diatom chloroplasts and provides explanations regarding the ecological success of this group of organisms and tips on how best to use the biology of these organisms in biotechnology.
The work mentioned the possibility of using diatom in biotechnology, but this fragment was poorly developed without providing examples of species and practical application. Nothing has been mentioned about the use of diatoms in agriculture and their important role. In the context of work, these data should be supplemented.
>> We thank the reviewer for their comments. We have added contextual statements between lines 55 and 60 that “Diatoms can be applied as aquafeeds for fish … in next-generation “circular” aquaculture techniques with increased capacity and reduced environmental impact . “, to clarify this important point. We have also provided specific examples of species usable for particular biotechnological applications, e.g. the use of benthic pennate diatoms for the production of UV-resistant photovoltaic cells; and Navicula glaciei as a source of ice-binding proteins for blood cryopreservation. While we feel that this is a fascinating and important topic, the broad scope of our paper limits the depth into which we can further explore this topic.
The authors may consider the unification of units in Figure 1. This will facilitate the possibility of comparing the given values for CO2 and O2 content in%.
>> We have chosen ppm and % respectively for CO2, and O2, as these are internationally accepted units for each (the 400 ppm CO2 threshold, and 21% atmospheric O2, ibid); and the effective concentrations of each compound are separated by over two orders of magnitude.
Fig. 3, 4 Descriptions in the drawings should be improved because they are of low quality.
>> We have traced over each metabolic pathway shown in fig. 3A with bold lines, to render them more visible; and have replaced the heatmap in fig 4 with wheel charts, which should be easier to visually follow (see comments to reviewer 2).
The captions in fig. 5 are illegible. The quality of the descriptions should be improved.
>> We apologise for the poor legibility in fig. 5B. We have resized all of the text within this panel to a minimum of Arial point 6, and have removed extraneous text (numerical support values, and branches composed of one species only).
Reviewer 2 Report
This contribution seeks to identify metabolic innovations in the chloroplast proteome that may have contributed to the extraordinary success and diversification of diatoms, further mining the database developed by Dorrell et al 2017 supporting there observations on evolution and diversification of the strameopile chloroplast proteome. Here the authors identify novelty in nuclear encoded plastid targeted enzymes that distinguish diatom plastid proteomes from other stramenopiles and to identify metabolic novelty between diatom species that help characterize taxa for bioproducts like lipid/oil production. While Ln 1 indicates this is a review article, and indeed text through Ln 190 provide a nice concise review of diatom evolution and global diversification, the manuscript is much more, with the remainder of the paper providing bioinformatic analysis and experimental support for plastid targeting of novel gene fusions in Fistulifera solaris making it attractive for bioproduct optimization. This reviewer just learned that the ms is a special edition contribution so perhaps a review designation is appropriate.
While I found the paper well written over all, the figures as provided in the review copy just don't do it justice, being very hard to read at current scaling. This is particularly true for Fig 3 A, Fig 4 and Fig 5. The reviewer had difficulty interpreting Fig 4 in particular, with no clear indication of color scaling, and gray scale interpretation. Perhaps split apart into two tables as the 'any protein' rows are hard to see with current gray scaling.
Figure 3 and 5 legends were truncated on the review copy, so need to be re-checked for completeness
Other edits:
Ln 132 through another algal groupn (e.g. .....
Author Response
This contribution seeks to identify metabolic innovations in the chloroplast proteome that may have contributed to the extraordinary success and diversification of diatoms, further mining the database developed by Dorrell et al 2017 supporting there observations on evolution and diversification of the strameopile chloroplast proteome. Here the authors identify novelty in nuclear encoded plastid targeted enzymes that distinguish diatom plastid proteomes from other stramenopiles and to identify metabolic novelty between diatom species that help characterize taxa for bioproducts like lipid/oil production. While Ln 1 indicates this is a review article, and indeed text through Ln 190 provide a nice concise review of diatom evolution and global diversification, the manuscript is much more, with the remainder of the paper providing bioinformatic analysis and experimental support for plastid targeting of novel gene fusions in Fistulifera solaris making it attractive for bioproduct optimization. This reviewer just learned that the ms is a special edition contribution so perhaps a review designation is appropriate.
While I found the paper well written over all, the figures as provided in the review copy just don't do it justice, being very hard to read at current scaling. This is particularly true for Fig 3 A, Fig 4 and Fig 5.
>> As described in the response to reviewer 1, we have traced over fig. 3A, and resized all of the text in fig. 5B, to render them more legible at A4 size.
The reviewer had difficulty interpreting Fig 4 in particular, with no clear indication of color scaling, and gray scale interpretation. Perhaps split apart into two tables as the 'any protein' rows are hard to see with current gray scaling.
>> We have replaced the heatmap version of fig. 4 with a series of wheel charts, showing (on the inner axis) the number of chloroplast-targeted copies and (on the outer axis) the number of species with orthologoues of particular proteins of interest, which we believe should be more legible to the casual reader. We regret that this figure is particularly information-dense and will benefit from detailed inspection from the reader in any presentation format.
Figure 3 and 5 legends were truncated on the review copy, so need to be re-checked for completeness
>> We confirm that the figure legends are not truncated in the resubmission copy.
Other edits:
Ln 132 through another algal groupn (e.g. .....
>> We thank the reviewer for their comment, and have corrected this error.